# Efficacy and Safety of Ethylene-Vinyl Alcohol (EVOH) Copolymer-Based Non-Adhesive Liquid Embolic Agents (NALEAs) in Transcatheter Arterial Embolization (TAE) of Acute Non-Neurovascular Bleeding: A Multicenter Retrospective Cohort Study

**DOI:** 10.3390/medicina59040710

**Published:** 2023-04-04

**Authors:** Roberto Minici, Massimo Venturini, Federico Fontana, Giuseppe Guzzardi, Armando Pingitore, Filippo Piacentino, Raffaele Serra, Andrea Coppola, Rita Santoro, Domenico Laganà

**Affiliations:** 1Radiology Unit, Pugliese-Ciaccio Hospital, 88100 Catanzaro, Italy; 2Diagnostic and Interventional Radiology Unit, ASST Settelaghi, Insubria University, Varese 21100, Italy; 3School of Medicine and Surgery, Insubria University, 21100 Varese, Italy; 4Radiology Unit, Maggiore della Carità University Hospital, 28100 Novara, Italy; 5Vascular Surgery Unit, Department of Medical and Surgical Sciences, Magna Graecia University of Catanzaro, University Hospital Mater Domini, 88100 Catanzaro, Italy; 6Haemophilia and Thrombosis Center, Pugliese-Ciaccio Hospital, 88100 Catanzaro, Italy; 7Radiology Unit, Department of Experimental and Clinical Medicine, Magna Graecia University of Catanzaro, University Hospital Mater Domini, 88100 Catanzaro, Italy

**Keywords:** EVOH, Onyx, Squid, TAE, embolization, bleeding, hemorrhage, NALEA, liquid embolics, endovascular

## Abstract

*Background and Objectives:* Transcatheter arterial embolization (TAE) is part of the daily practice of most interventional radiologists worldwide. The ideal liquid embolic agent is far from being identified. Non-adhesive liquid embolic agents (NALEA) harden from the outside to the inside, resulting in deep penetration, known as “magma-like” progression, which permits a more distal embolization with good control of the embolic material. This multicenter retrospective cohort study aims to assess the efficacy, feasibility and safety of transcatheter arterial embolization (TAE) with ethylene-vinyl alcohol (EVOH)-based NALEAs (Onyx and Squid) in acute bleeding outside of the neurovascular area. *Materials and Methods:* This study is a multicenter analysis of retrospectively collected data of consecutive patients who had undergone, from January 2015 to December 2022, transcatheter arterial embolization with non-adhesive EVOH-based agents in the setting of acute non-neurovascular bleeding. *Results:* Fifty-three patients underwent transcatheter arterial embolization for acute non-neurovascular bleeding. Eight (15.1%) procedures were performed in patients with coagulopathy. The most used concentration of EVOH-based NALEAs was 34 (i.e., 8%), with a mean dose of 0.5 (±0.3) mL. The mean CT-to-groin time, the mean procedure time, the mean CT-to-embolization time and the mean fluoroscopy time were 22.9 (±12.4) min, 27.5 (±7) min, 50.3 (±13.1) min and 7.5 (±2.8) min, respectively. Technical success was achieved in all cases with a 96.2% clinical success rate. Complications were recorded in six (11.3%) patients. No statistically significant differences were observed between the group of patients with coagulopathy and the group of patients without coagulopathy in terms of efficacy and safety endpoints. *Conclusions:* Transcatheter arterial embolization (TAE) performed with non-adhesive EVOH-based embolic agents is an effective, feasible and safe strategy for the management of acute non-neurovascular bleeding, even in the subgroup of patients with coagulopathy.

## 1. Introduction

Transcatheter arterial embolization (TAE) is part of the daily practice of most interventional radiologists worldwide and is now one of the cornerstones of acute bleeding management, in accordance with numerous international guidelines [1,2,3]. The endovascular management of acute bleeding is effective and safe in many areas of the body, both in the intracranial and extracranial districts [4,5,6,7,8,9,10,11,12,13,14,15,16]. Despite the consolidation of TAE in clinical practice, considerable data heterogeneity remains due to several variables, such as the causes of bleeding, the clinical and coagulation status of the patient, the embolization technique and the embolic agent used. Therefore, embolotherapy for acute bleeds is an active area of research.

The ideal liquid embolic agent is far from being identified. It should be quickly effective; reach and fill the targeted vessels without migration; easy to prepare; highly radiopaque and, therefore, clearly visible; controllable during administration; biocompatible; and cost-effective [17]. Non-adhesive liquid embolic agents (NALEA) have non-adhesive properties because they harden from the outside to the inside, resulting in deep penetration, known as “magma-like” progression, which permits a more distal embolization with good control of the embolic material [18]. NALEAs typically consist of a copolymer bonded to a high atomic weight radiopaque compound that allows them to be visualized under X-rays [19]. Among copolymers, the most frequently used is ethylene-vinyl alcohol (EVOH). Onyx (Medtronic, Irvine, CA, USA) and Squid (Balt, Montmorency, France) are EVOH-based NALEAs consisting of EVOH copolymer, dimethyl-sulfoxide (DMSO) and tantalum powder added for radiopacity. Compared to Onyx, Squid was introduced more recently, is manufactured with a viscosity of 12 centistokes (cSt) (i.e., a measure of kinematic viscosity) and has a smaller micronized tantalum powder aiming to enhance the homogeneity in radiopacity and improve visibility during longer injection times [18,19,20,21,22,23]. Onyx was first described in the 1990s as an embolizing agent for the treatment of intracranial arteriovenous malformations (AVMs). Subsequently, its use was also consolidated in the extracranial district, mainly for AVMs, bronchial and pulmonary arteries, endoleaks, visceral aneurysms and pseudoaneurysms, hypervascular tumors and venous embolizations [24,25,26,27,28,29,30,31,32]. Squid has been used mainly in the cerebral district [23,33] with some recent applications in the extracranial district [34,35,36].

To date, few studies have selectively investigated non-adhesive EVOH-based agents in the setting of acute non-neurovascular bleeding, with a prevalence of case reports and small case series [17,28,30,35,37,38,39,40,41,42,43,44,45,46,47,48,49,50,51,52,53,54,55,56,57,58,59,60,61,62,63,64].

This multicenter retrospective cohort study aims to assess the efficacy, feasibility and safety of transcatheter arterial embolization (TAE) with ethylene-vinyl alcohol (EVOH)-based NALEAs (Onyx and Squid) in acute bleeding outside of the neurovascular area.

## 2. Materials and Methods

### 2.1. Study Design

This study is a multicenter (Pugliese-Ciaccio Hospital, Catanzaro, Italy; Circolo Hospital, Varese, Italy; Maggiore della Carità University Hospital, Novara, Italy; Mater Domini University Hospital, Catanzaro, Italy) analysis of retrospectively collected data of consecutive patients who had undergone, from January 2015 to December 2022, transcatheter arterial embolization with non-adhesive EVOH-based agents in the setting of acute non-neurovascular bleeding (Figure 1, Figure 2 and Figure 3). Inclusion criteria include (I) TAE performed due to acute non-neurovascular bleeding, according to the indications provided by the Society of Interventional Radiology (SIR) guidelines for percutaneous transcatheter embolization [3]; (II) use of a non-adhesive EVOH-based agent (Onyx or Squid) as the main embolic agent; (III) age of at least 18 years; (IV) evaluation by a multidisciplinary team of surgeons, interventional radiologists and anesthesiologists. The exclusion criteria include (I) pregnant or breastfeeding women; (II) platelet count <20,000/μL and refusal of transfusion, according to SIR guidelines for low bleeding risk procedures that require arterial access [65]; (III) international normalized ratio (INR) greater than or equal to 1.8 for femoral access or greater than or equal to 2.2 for radial access, according to SIR guidelines for low bleeding risk procedures that require arterial access [65]; (IV) hypersensitivity to EVOH copolymer and/or DMSO solvent; (V) bleeding sustained by the internal carotid artery or its branches.

Ethics committee approval was not required due to the retrospective nature of the study. The study has been conducted in accordance with the Helsinki Declaration. All patients signed a written informed consent before receiving the endovascular treatment.

### 2.2. Treatment

All patients underwent pretreatment evaluation with CT angiography (CTA), except in special cases, according to international guidelines (e.g., hemodynamically unstable patients with pelvic trauma, in accordance with WSES guidelines [66]). The endovascular procedure was performed in dedicated angiographic suites by an experienced interventional radiologist. Diagnostic angiography has always preceded the superselective catheterization of the bleeding/pseudoaneurysm-feeding arteries. NALEAs were prepared according to the instructions for use and injected under fluoroscopic guidance via a DMSO-compatible microcatheter. The dead space of the microcatheter, including the luer lock hub at its proximal end, was filled with DMSO to prevent EVOH precipitation. The microcatheter was never reused or washed with DMSO at the end of the injection. Assessment of technical success and nontarget embolization were performed by postembolization angiography, taking into account the possible collateral circulation based on the anatomical site of the bleeding. The anesthesiologist performed sedation during the embolization to improve patient comfort and provided analgesic therapy after the procedure. All patients underwent clinical evaluation and follow-up imaging before the hospital discharge and 1 month after TAE.

### 2.3. Outcomes and Definitions

The primary efficacy endpoint was the rate of technical success. The clinical success rate was selected as a secondary efficacy endpoint. The primary safety endpoint was the rate of complications. The nontarget embolization rate and the rate of major complications graded according to the 2003 SIR classification [67] were selected as secondary safety endpoints. The primary feasibility endpoint was the procedure time. 

Unless otherwise specified, reporting standards of the Society for Interventional Radiology for percutaneous transcatheter embolization have been used [68]. The coagulopathy subgroup was defined as in Loffroy et al. [69]: INR greater than 1.5, partial thromboplastin time longer than 45 s or platelet count less than 80,000/mm^3^. CT-to-groin time, procedure time and CT-to-embolization time were calculated taking into account the times indicated in the CT report and the surgical operative log. Complications linked to TAE were graded according to the 2017 SIR classification [70], the 2003 SIR classification [67] and the CIRSE classification [71].

### 2.4. Statistical Analysis

Data were maintained in an Excel spreadsheet (version 16.67—Microsoft Inc, Redmond, DC, USA) and the statistical analyses were performed on an intention-to-treat basis, using SPSS software (SPSS, version 22 for Windows; SPSS Inc., Chicago, IL, USA) and R/R Studio software (R version 4.2.2—R Studio Team, Boston, MA, USA). The analyses were based on the Modified Intention-To-Treat population, defined as all randomized patients who received at least one embolization treatment [72,73]. Kolmogorov–Smirnov test and Shapiro–Wilk test were used to verify the normality assumption of data. Categorical data are presented as frequency (percentage value). Continuous normally distributed data are presented as mean ± standard deviation. Continuous non-normally distributed data are presented as median (first–third quartile). The unpaired Student’s t-test was used to assess statistical differences for continuous normally distributed data, while categorical and continuous non-normally distributed data were assessed using the Chi-squared/Fisher’s exact tests and the Mann–Whitney test, respectively. A *p*-value of <0.05 was considered statistically significant for the aforementioned tests.

## 3. Results

During the study interval (January 2015–December 2022), a total of 53 patients underwent transcatheter arterial embolization for acute non-neurovascular bleeding. Ten (18.9%) procedures were performed in patients with COVID-19 and eight (15.1%) procedures were performed in patients with coagulopathy. CT angiography was performed in 47 (88.7%) cases and in all these cases bleeding was detected. The mean hematoma volume measured on CT was 231.3 (±298.6) mL. Twenty-one (39.6%) patients were on antiplatelet therapy and thirteen (24.5%) patients were on anticoagulant therapy. A total of 30 (56.6%) patients were on antiplatelet or anticoagulant therapy and only 2 (3.8%) patients were on both antiplatelet and anticoagulant therapy. Details are given in Table 1.

A total of 53 transcatheter arterial embolizations were performed. In three cases, no bleeding was detected on X-ray angiography (XA); therefore, a blind embolization was performed (i.e., embolization of three gastroduodenal arteries guided by the clip released by the endoscopist). The most common bleeding site was the abdomen (34%), followed by the pelvis (28.4%). The primary cause of bleeding was trauma (58.5%). The most used concentration of EVOH-based NALEAs was 34 (i.e., 8%), with a mean dose of 0.5 (±0.3) mL. No additional embolic agents were needed. The mean volume of iodinated contrast media used during embolizations was 35.2 (±9.4) mL. The mean volume of contrast to creatinine clearance ratio was 0.7 (±0.6). The most common vascular access site was the common femoral artery (75.5%). The mean CT-to-groin time, the mean procedure time, the mean CT-to-embolization time and the mean fluoroscopy time were 22.9 (±12.4) min, 27.5 (±7) min, 50.3 (±13.1) min and 7.5 (±2.8) min, respectively. Radiation exposure expressed by cumulative air kerma and total dose area product was recorded as 160 (±58.3) mGy and 25.1 (±9.3) Gy/cm^2^ respectively.

Procedure data are detailed in Table 2.

Technical success was achieved in all cases with a 96.2% clinical success rate related to two cases of rebleeding. Only one case of nontarget embolization was observed and with no clinical consequences. Complications were recorded in six (11.3%) patients. The rate of vascular access site complications (VASCs) was 3.8%, which was related to only two cases of hematoma. According to the 2017 SIR classification for complications [70], four (7.5%) patients experienced a procedure-related grade 2 event (two access site hematomas, one post-embolization syndrome, and one abscess), one (1.9%) patient experienced a procedure-related grade 3 event (acute pancreatitis) and one (1.9%) patient experienced a procedure-related grade 5 event. The 30-day mortality rate was 1.9%, which was related to a case of deep vein thrombosis in the same limb undergoing compression bandaging, resulting in pulmonary embolism and patient death. 

Details are given in Table 3.

No statistically significant differences were observed between the group of patients with coagulopathy and the group of patients without coagulopathy in terms of the prevalence of COVID-19, platelet count, intake of antiplatelet or anticoagulant therapy, embolic agent dose, procedure time, fluoroscopy time, technical success, clinical success (Figure 4), rebleeding or complications (Figure 5).

A comparison of data between patients with and without coagulopathy is reported in Table 4.

## 4. Discussion

In this multicenter retrospective cohort study, non-adhesive EVOH-based agents (Onyx and Squid) have been effective in the embolization of acute non-neurovascular bleeds. Technical success was achieved in all cases with a clinical success rate of 96.2%. These results are comparable with those reported from other studies evaluating EVOH-based NALEAs in acute hemorrhage [28,30,62,64,74]. Lucatelli et al. reported a clinical success rate of 97.15% in peripheral embolizations, including arterial bleedings, performed using Phil (Precipitating Hydrophobic Injectable Liquid, MicroVention Terumo, Tustin, CA, USA), a novel non-EVOH-based NALEA [11]. Another recently introduced NALEA is EASYX, which is composed of a polyvinyl alcohol (PVA) polymer covalently linked to radiopaque iodine groups and has shown a technical success of 98% in a recent study by Sapoval et al. [6]. Thus, the efficacy of EVOH-based NALEAs in the setting of acute non-neurovascular bleeding is very high and comparable to that of other NALEAs. Furthermore, our results exceed the Society of Interventional Radiology’s suggested thresholds and reference values for percutaneous embolizations, which account for studies also performed with solid and liquid non-adhesive embolic agents [68]. 

This finding could be explained by some technical advantages related to the use of EVOH-based NALEAs. Firstly, the magma-like progression allows the embolic agent to be carried through small-caliber arteries, reaching embolization targets that are significantly far from the microcatheter tip [18,55]. If the bleeding comes from a single vessel and the vessel caliber is not too small, the embolization with coils is very effective. Conversely, the use of coils is less effective in cases of multiple bleeding sources, small caliber vessels that are difficult to catheterize and backdoor bleeding [75,76]. Secondly, EVOH-based NALEAs guarantee a safe, rapid and effective embolization via polymerization, and, therefore, an embolization of a mechanical nature that does not require the activation of coagulation. This feature is of particular interest in patients with coagulation disorders [18,55] and is confirmed by the clinical data we have collected. Indeed, we did not observe statistically significant differences in terms of technical success and clinical success in the subgroup of patients with coagulopathy. It is worth noting that the subgroup of patients with coagulopathy showed a statistically significant prolongation of INR. Interestingly, the use of coils as the sole embolic agent and the presence of coagulopathy have previously been shown to be independent predictors of early rebleeding [69]. Coil embolization is less effective in patients with coagulopathy and liquid embolics do not seem to suffer from this limitation [64,69,77,78,79,80]. Finally, it is worth emphasizing that an alteration of the clotting function is very frequent in trauma patients, which is also due to consumption coagulopathy [81]. 

Our investigation demonstrates that TAE for acute non-neurovascular bleeds is a safe procedure when performed with non-adhesive EVOH-based agents. The safety outcomes are similar to other investigations in the field of endovascular treatments and TAEs [6,28,30,48,52,63,68,74,82,83,84,85,86,87,88]. In a recent systematic review on the peripheral embolization of acute hemorrhage performed with Onyx, Kolber et al. [74] observed a 9.2% mortality rate at 30 days, a 7.6% rebleeding rate and a 3.1% major complication rate. Né et al. [28] reported a 4% rebleeding rate, that 20% of patients suffered pain during injection and no major complications in a series of fifty patients undergoing Onyx embolization for peripheral applications. No procedure-related complications or bowel ischemia was observed by Lenhart et al. [48] or Sun et al. [63] in 15 and 9 patients, respectively, undergoing TAE with Onyx for acute gastrointestinal hemorrhage. Urbano et al. [52] reported only one case of Onyx reflux without clinical sequelae in 31 patients with lower gastrointestinal bleeding. Regine et al. [30] observed a 15.4% minor complication rate (fever and pain) in their series on the embolization of traumatic and non-traumatic peripheral vascular lesions with Onyx. Sapoval et al. reported 3 major device-related complications (6%) in their case series of 50 peripheral embolizations performed with an EASYX liquid embolic agent [6]. In a recent systematic review and meta-analysis, Kim et al. [82] found a complication rate of 5.4% for upper gastrointestinal bleeding embolizations and 4% for lower gastrointestinal bleeding embolizations performed with N-butyl cyanoacrylate (NBCA).

Unlike other liquid embolics such as NBCA, EVOH does not adhere to the endothelial and microcatheter walls and, thus, minimizes the inflammation of the endothelium and the risk of microcatheter obstruction and entrapment [28,64,74]. According to Khalil et al. [45], EVOH should be preferred when performing the embolization of damaged vessels (e.g., in aspergillomas, tuberculous and cancerous lesions) because it deploys without applying a radial force to vessel walls, contrary to the coils and vascular plugs, which can rupture a fragile vessel (Figure 6). Several reports have highlighted a risk of rebleeding with microspheres and coils [17,38,39,46]. Microspheres may be shunted through collaterals or acquired fistulae, and multiple coils may be difficult to deploy throughout tortuous vessels, thus increasing safety concerns [38,46]. Thanks to its non-adhesive properties and slow polymerization, EVOH adapts to even the most tortuous anatomies and forms a cast of the vessels it enters. It can also reach targets very distant from the microcatheter because of its low viscosity formulations [40,44,74,89]. In high-flow lesions, where particles and NBCA are difficult to control, the high viscosity formulation and magma-like consistency of EVOH were preferred to decrease the risk of nontarget embolization and organ infarction [51,52]. Adhering to a slow injection rate that does not exceed the recommended limit of 0.3 mL/min limits the risk of reflux and, consequently, the risk of nontarget embolization, allowing better control of embolic agents’ delivery (Figure 7) [45,51,52]. Interestingly, it is also possible to use the reflux-hold-reinjection technique, which consists of safely interrupting the injection and resuming it after the formation of a first cast to reduce the risk of reflux [40]. Our data confirm that the use of EVOH is highly safe in embolization procedures provided that the correct injection technique is followed. This confirmation allows us to make the most of its chemical–physical characteristics.

However, some limitations of EVOH-based NALEAs should be considered. Firstly, DMSO is a byproduct of paper manufacturing [90], requires the use of a DMSO-compatible microcatheter and is toxic to blood vessels, causing vasospasm, pain and unusual reflexes such as the trigeminocardiac reflex [74,91,92,93,94]. Few cases of acute respiratory distress syndrome during neurovascular interventions with Onyx have been described and an immune-mediated reaction during the pulmonary removal of the DMSO has been hypothesized [95,96]. These effects may be limited by slowing DMSO release during the initial EVOH injection into the microcatheter, limiting subsequent microcatheter flushes with DMSO, and providing adequate patient sedation [11,40,48,74,97]. Secondly, DMSO is volatile and is excreted through respiration and sweat, so the patient may smell garlic on their breath and smell it for about two days after the procedure [28,97]. Thirdly, tantalum powder determines the radiopacity of EVOH-based NALEAs, ensuring excellent visibility during the procedure, but causing beam hardening artefacts on follow-up CT exams. Therefore, both Onyx and Squid feature reduced tantalum formulations, called Onyx L and Squid LD, respectively, to reduce beam hardening artifacts [47,98]. No tantalum-associated artefacts are described on magnetic resonance imaging and tantalum produces a hypointense signal on both T1-weighted and T2-weighted sequences [99]. In addition, if subcutaneous or submucosal vessels are embolized, tantalum can be deposited in the tissues and cause permanent visible tattoos on the skin or mucous membranes [100]. If a long segment of a microcatheter is filled with EVOH within a tortuous vessel, the mechanical non-adhesive trapping of the microcatheter might occur [101]. In cases such as this, the gradual traction of the microcatheter until its eventual release has been suggested [102]. Patients should be fully informed about these rare, but possible, potential adverse effects so that they can express informed consent to the proposed treatment. 

EVOH needs to be shaken for 20 min to obtain a homogeneous mixture of EVOH and tantalum powder and achieve sufficient radiopacity, thus limiting its use in the case of emergency [28,46,97]. In our institute, this issue has been addressed by instructing staff to start shaking the vials with a Vortex shaking mixer when activating the cath lab for an embolization procedure. The collected data confirm the feasibility of our protocol; indeed, the mean procedure time is similar to that of other studies on TAE [11,103,104,105]. Therefore, TAE using EVOH-based NALEAs appears to be a feasible strategy for the management of acute non-neurovascular bleeds in a typically time-sensitive scenario. 

The use of EVOH-based NALEA in peripheral embolization may be limited by the high cost, according to multiple authors [17,38,47,92]. While the costs are higher than those of NBCA and pushable coils, there does not appear to be a cost difference with detachable coils. In Europe, a vial of EVOH costs around EUR 600–800, while a detachable coil costs around EUR 200–800 [64]. One vial of EVOH is often sufficient for a peripheral embolization procedure, while very rarely a single detachable coil may be sufficient [64]. Thus, a cost analysis reveals that EVOH embolization is on average no more expensive than detachable coil embolization.

Finally, as stated by Tipaldi et al. [64], the use of liquid embolics requires a longer learning curve than the coils, but the use of EVOH-based NALEAs is probably more “user-friendly” than the NBCA due to the lower risk of insidious adverse events, such as the gluing of the tip of the microcatheter, and the greater control that the operator has over the path of the embolic agent.

The limitations of the study include the lack of a control group, the retrospectivity of the analysis, the heterogeneity of indications, the short-term follow-up and the scarcity of data in the literature that is necessary to evaluate the congruence and the consistency of the data presented.

## 5. Conclusions

To the best of our knowledge, no multicenter cohort studies have, so far, investigated the efficacy, feasibility and safety profile of EVOH-based NALEAs (Onyx and Squid) for acute non-neurovascular hemorrhage.

Hence, the results of the current investigation demonstrate that transcatheter arterial embolization (TAE) performed with non-adhesive EVOH-based embolic agents has efficacy, feasibility and safety comparable to TAE performed with other embolic agents. Interestingly, the efficacy and the safety profile were similar in the subgroup of patients with coagulopathy.

Full knowledge of the technical details of their use is essential to make the most of these embolic agents and minimize their risks.

Larger, randomized and controlled trials are desirable to confirm our data.

## Figures and Tables

**Figure 1 medicina-59-00710-f001:**
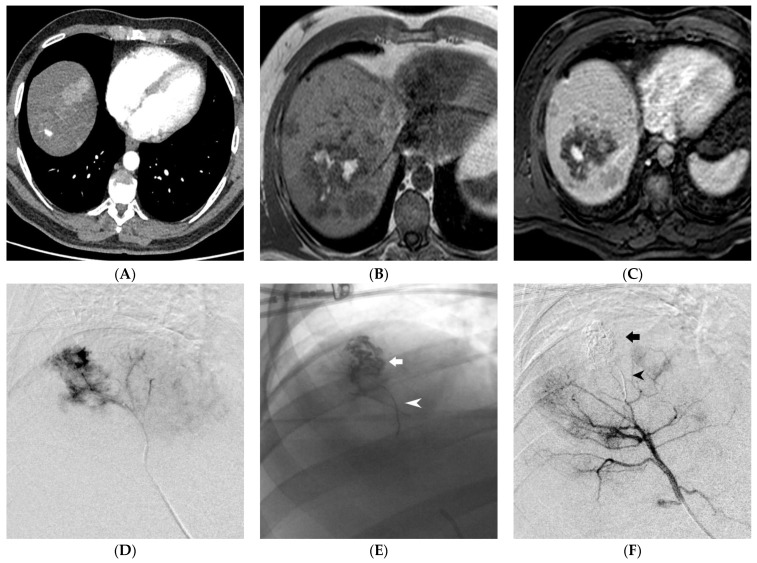
Ruptured Hepatocellular Carcinoma (HCC) within the hepatic parenchyma. Contrast-enhanced CT, arterial phase: evidence of 1 cm pseudoaneurysm within an HCC nodule (**A**). MRI, T1 Fast Field Echo In-Phase sequence: hyperintense material (blood) within a large HCC nodule of the right lobe (**B**). Gd-BOPTA-enhanced MRI, arterial phase: intranodular contrast blush confirmed (**C**). Superselective digital subtraction angiography of the S7 hepatic artery branch demonstrating contrast leakage into the ruptured HCC (**D**). Effective embolization of the HCC nodule (arrow) and the parent artery (arrowhead) using EVOH copolymer (**E**,**F**).

**Figure 2 medicina-59-00710-f002:**
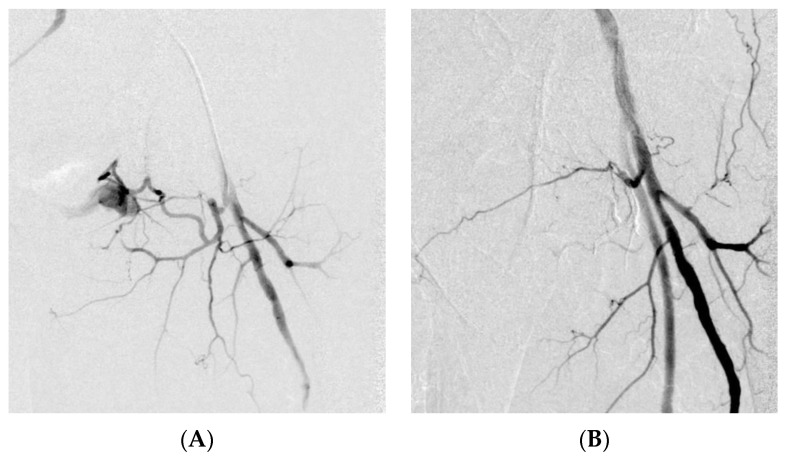
Road traffic collision. Selective digital subtraction angiography showing injury of an ascending branch of the medial circumflex artery with pseudoaneurysm formation (**A**). Digital subtraction angiography demonstrates the filling properties of the EVOH copolymer, resulting in effective embolization (**B**).

**Figure 3 medicina-59-00710-f003:**
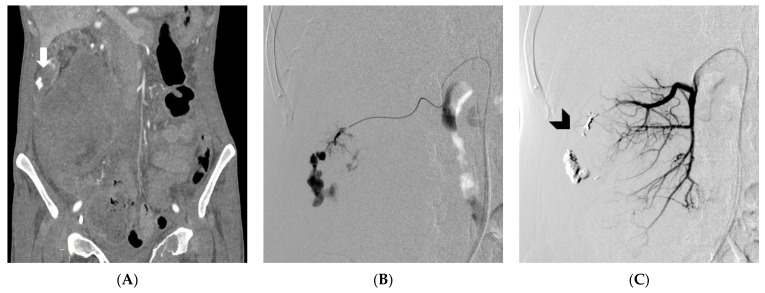
CT angiography of spontaneous retroperitoneal bleeding due to ruptured renal tumor pseudoaneurysm (arrow) (**A**). Digital subtraction angiography confirming ruptured pseudoaneurysm arising from a tumor feeding artery (**B**). Digital subtraction angiography showing effective embolization with EVOH copolymer cast (arrowhead) (**C**).

**Figure 4 medicina-59-00710-f004:**
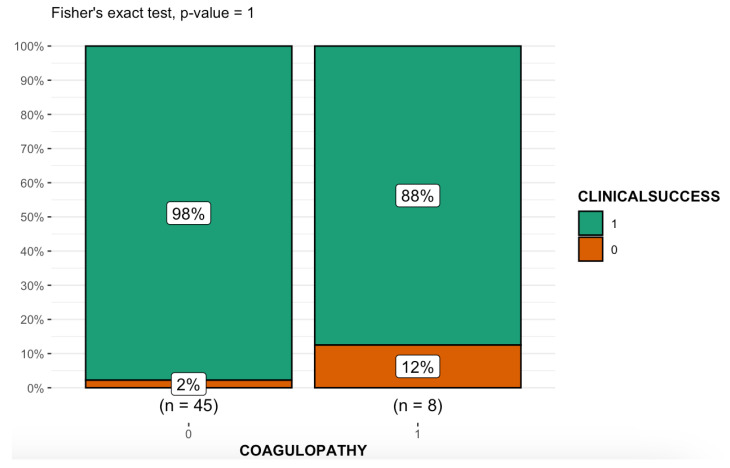
Boxplot representing the relationship between the two categorical variables, “Coagulopathy” and “Clinical Success”, analyzed using Fisher’s exact test.

**Figure 5 medicina-59-00710-f005:**
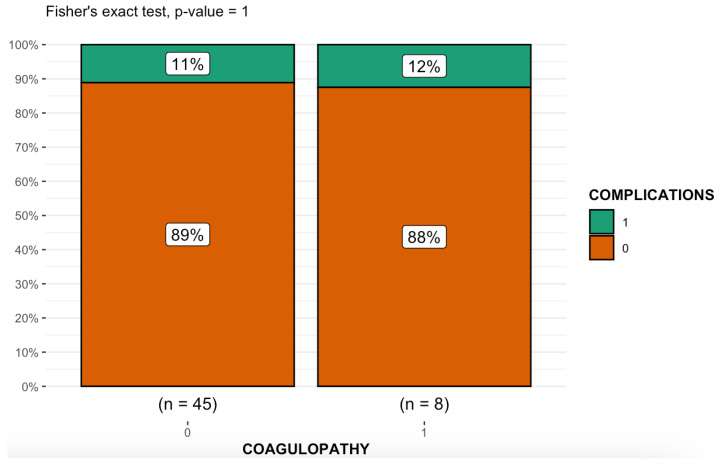
Boxplot representing the relationship between the two categorical variables, “Coagulopathy” and “Complications”, analyzed using Fisher’s exact test.

**Figure 6 medicina-59-00710-f006:**
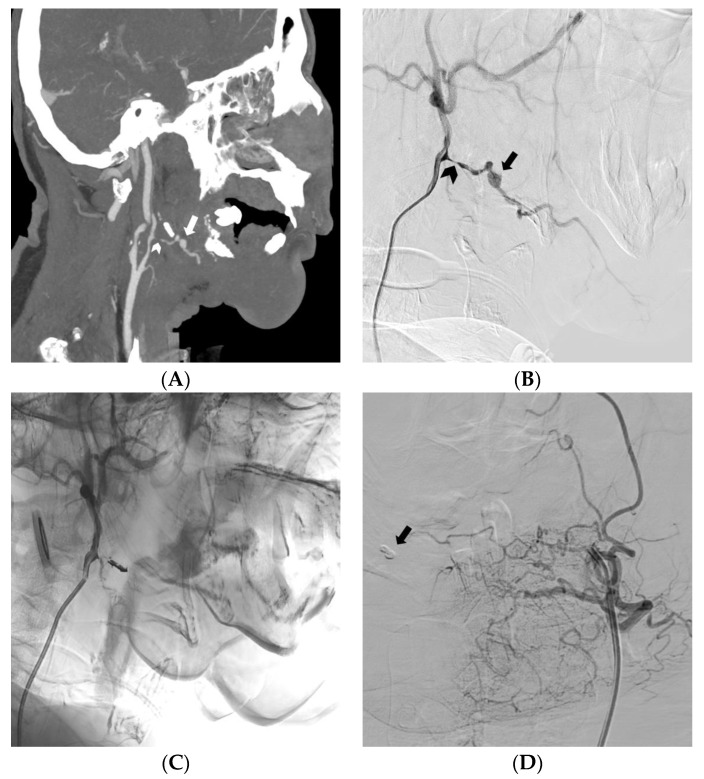
Locally advanced squamous cell carcinoma of the tongue. CT angiography with maximum intensity projection reformatting in the sagittal plane (**A**) and digital subtraction angiography of the external carotid artery (**B**). Massive oral bleeding from a pseudoaneurysm (arrow) of a heavily eroded (arrowhead) right lingual artery (**A**,**B**). Angiogram demonstrating effective embolization of the lingual artery with EVOH copolymer (**C**). Contralateral digital subtraction angiography excluding retrograde bleeding; the onyx cast can be noted (arrow) (**D**).

**Figure 7 medicina-59-00710-f007:**
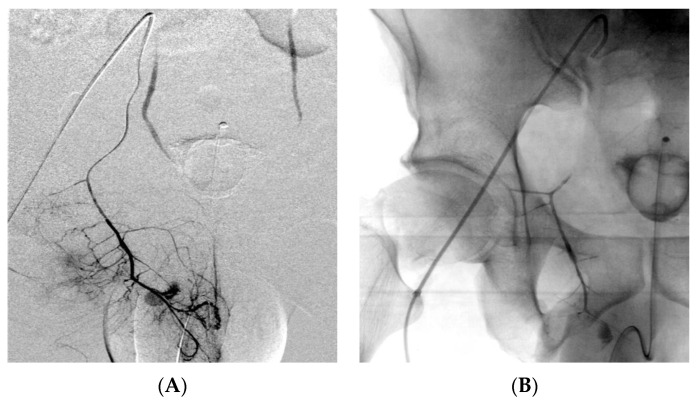
Pedestrian versus motorcycle accident. Selective digital subtraction angiography showing injury of the internal pudendal artery (**A**). The plain radiograph demonstrates the distribution of 1 mL of EVOH copolymer with intrinsically radiopaque tantalum powder (an injection that was performed too quickly resulted in reflux within the parent artery and side branches with no clinical consequences) (**B**).

**Table 1 medicina-59-00710-t001:** Population data.

Variables	All Patients (*n* = 53)
Age (years)	66.2 (±15)
Sex (M/F)	34 (64.2%)/19 (35.8%)
COVID-19	10 (18.9%)
eGFR (mL/min)	68 (±25)
CKD Stage	2 (1–3)
INR	1.2 (±0.3)
aPTT	35.8 (±4.8)
Platelet count (No. ×10^3^/μL)	386.1 (±73.4)
Coagulopathy	8 (15.1%)
Hemoglobin (g/dL)	7.7 (±0.7)
CT angiography execution	47 (88.7%)
Bleeding on CT angiography	47 (88.7%)
Hematoma volume (mL)	231.3 (±298.6)
Antiplatelet therapy - Single list - Dual	21 (39.6%) 15 (28.3%) 6 (11.3%)
Anticoagulant therapy	13 (24.5%)
Antiplatelet AND anticoagulant therapy	2 (3.8%)
Antiplatelet OR anticoagulant therapy	30 (56.6%)

**Table 2 medicina-59-00710-t002:** Procedure data.

Variables	All Patients (*n* = 53)
Bleeding on XA	50 (94.3%)
Blind embolization	3 (5.7%)
Site of bleeding - Pelvic - Upper GI - Lower GI - Abdomen - Thorax - Neck - Limbs	15 (28.4%) 4 (7.5%) 2 (3.8%) 18 (34%) 8 (15%) 2 (3.8%) 4 (7.5%)
Number of embolized vessels	1.4 (±0.8)
Cause of bleeding - Trauma - Spontaneous - Others (tumors, diverticula, ulcers, etc.)	31 (58.5%) 5 (9.4%) 16 (32.1%)
Embolic agent concentration - 12 - 18 - 20 - 34	4 (7.5%) 6 (11.3%) 20 (37.7%) 23 (43.5%)
Embolic agent dose (mL)	0.5 (±0.3)
Use of an additional embolic agent	0 (0%)
Intraoperative unfractionated heparin (IU)	528.3 (±952.8)
Intraoperative contrast medium (mL)	35.2 (±9.4)
Volume of contrast to creatinine clearance ratio	0.7 (±0.6)
Vascular access site - Femoral - Radial - Brachial	40 (75.5%) 11 (20.7%) 2 (3.8%)
Sheath diameter, 4F/5F/6F/≥7F	6 (11.3%)/44 (83.0%)/3 (5.7%)/0 (0%)
Angiography injection technique, manual/powered	29 (54.7%)/24 (45.3%)
CT-to-groin time (min)	22.9 (±12.4)
Procedure time (min)	27.5 (±7)
CT-to-embolization time (min)	50.3 (±13.1)
Fluoroscopy time (min)	7.5 (±2.8)
Cumulative air kerma (mGy)	160 (±58.3)
Dose area product (DAP) (Gy/cm^2^)	25.1 (±9.3)

**Table 3 medicina-59-00710-t003:** Outcome data.

Variables	All Patients (*n* = 53)
Technical success	53 (100%)
Clinical success	51 (96.2%)
Vascular access site hemostasis - Manual compression - Vascular closure device	50 (94.3%) 3 (5.7%)
Units of packed red blood cells transfused per patient	1 (±0.6)
Rebleeding	2 (3.8%)
Nontarget embolization	1 (1.9%)
Complications	6 (11.3%)
Vascular access site complications (VASCs)	2 (3.8%)
Complications, according to SIR classifications - None - Minor (grade 1–2) - Major (grade 3–5)	47 (88.7%) 4 (7.5%) 2 (3.8%)
Complications, according to CIRSE classification - Grade 0 - Grade 3 - Grade 6	47 (88.7%) 5 (9.4%) 1 (1.9%)
Treatment required for complications - None - Medical - Interventional - Surgical	47 (88.7%) 5 (9.4%) 1 (1.9%) 0 (0%)
30-day mortality	1 (1.9%)

**Table 4 medicina-59-00710-t004:** Comparison of data between patients with and without coagulopathy.

Variables	Group 1 (*n*° = 8) Patients with Coagulopathy	Group 2 (*n*° = 45) Patients without Coagulopathy	*p* Value
COVID-19	2 (25%)	8 (17.8%)	0.6364
INR	1.6 (1.5–1.8)	1.1 (1–1.2)	0.0002
Platelet count (No. ×10^3^/μL)	393 (254–423)	394 (378–421)	0.4709
Antiplatelet OR anticoagulant therapy	4 (50%)	26 (57.8%)	0.7153
Embolic agent dose (mL)	0.5 (0.4–1)	0.4 (0.3–0.6)	0.3719
Procedure time (min)	30 (26–31)	27 (22–32)	0.4182
Fluoroscopy time (min)	7 (7–10)	7 (6–9)	0.2635
Technical success	8 (100%)	45 (100%)	1
Clinical success	7 (87.5%)	44 (97.8%)	0.2816
Rebleeding	7 (87.5%)	44 (97.8%)	0.2816
Complications	1 (12.5%)	5 (11.1%)	1

## Data Availability

The data presented in this study are available on request from the corresponding author. The data are not publicly available due to privacy issues.

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
