# Peer review of "Efficacy and Safety of Ethylene-Vinyl Alcohol (EVOH) Copolymer-Based Non-Adhesive Liquid Embolic Agents (NALEAs) in Transcatheter Arterial Embolization (TAE) of Acute Non-Neurovascular Bleeding: A Multicenter Retrospective Cohort Study"

_medicina, 2023, doi:10.3390/medicina59040710_

Round 1

Reviewer 1 Report

This is a retrospective monocentric study on 2 NALEAs on 53 procedures in 8 years. 

The article is well-written. The topic is well known and analyzed by authors. 

My 2 concerns are:

-101 references is huge for this type of article which is neither a review or a meta-analyses. The job could be done with 30 references

-the conclusion isn't supported by the results of this retrospective monocentric study on 53 participants. It should be rewritten and shaded.

Author Response

Dear Reviewer,

Thanks for your comments and suggestions.

We understand you’ve noted that there are only 53 participants, but please evaluate some points:

  • In the discussion section, we demonstrated that the efficacy and safety of TAE with NALEA are similar to that of TAE with other embolic agents;
  • Ours is a multicenter study, not a monocentric one;
  • It’s one of the largest cohort studies ever published on non-neurovascular acute bleeding treated with NALEAs.

We tried to modify the manuscript by shading the conclusion as you suggested.

Reviewer 2 Report

Article  „Efficacy and safety of ethylene-vinyl alcohol (EVOH) copoly- 2 mer-based non-adhesive liquid embolic agents (NALEAs) in 3 transcatheter arterial embolization (TAE) of acute non-neuro- 4 vascular bleeding: a multicenter retrospective cohort study“.

Very interesting, useful and relevant work.

In my opinion the manuscript is very interesting, useful and relevant.

I have only two recommendations:

1.       In the Discussion and Conclusion can be add the hypothesis about the statistical significant difference for International normalized ratio (INR) – Table 4. The information in row 132 is not enough to clearly explain the result for INR difference from Table 4.  The manuscript quality can be improved if the authors add this hypothesis.

2.       The authors can add an information about the fitness (moving) status of the 53 participants. It will be interesting how the physical activity (inactivity) level of participants will influence the obtained results. Hemorheological blood properties are of primary importance for the blood supply to tissues and organs in the body at rest and during sports training and physical exercises. In the work of Ivanov, 2022 was reflect the world experience of analysis of basic hemorheological changes in human blood, blood cells and plasma as a result of intense physical activity (physical exercises at different intensity, frequency, duration and load regimes).

In this aspect, scientific results are often contradictory, divergent and difficult to interpret.

Several major stimulatory effects of physical exercise on the basic systems of blood “hemostasis” have been described: enhanced blood coagulation (blood clotting), activated platelet aggregation and increased number of platelets entering the bloodstream presumably from the bone marrow, spleen, and lung, and enhanced fibrinolysis (degradation of fibrin fibers in eventual thrombogenesis) [Ivanov, I. Hemorheological Alterations and Physical Activity. Appl. Sci. 2022, 12, 10374. https:// doi.org/10.3390/app122010374; El-Sayed, M.S. Effects of Exercise on Blood Coagulation, Fibrinolysis and Platelet Aggregation. Sports Med. 1996, 22, 282–298.].

The increased (senile) physical inactivity leads to [Ivanov, I. Hemorheological Alterations and Physical Activity. Appl. Sci. 2022, 12, 10374. https:// doi.org/10.3390/app122010374]:

- oxidative and/or mechanical cell stress change;

- metabolic changes in cells and tissues (reduced pH, changes in tissue oxidation, accumulation of lactate);

- changes in respiratory lung functions;

- changes in the regulation and adaptation of vascular tone (for example, in changes in the mechanoreceptors of vascular endothelial cells, as well as in the synthesis of endothelial nitric oxide synthase).

Author Response

Dear Reviewer,

Thanks for your comments and suggestions.

Here are our answers:

  1. Fixed: the statistically significant difference in INR between subgroups has been stated in the discussion section.
  2. As it’s a retrospective multicenter study, we haven’t collected any data on the fitness status of the participants. Therefore, we apologize for being unable to address your suggestion.

Reviewer 3 Report

I congratulate the authors on their manuscript.

The main topic addressed by the research is transcatheter arterial embolization (TAE), which is part of the daily practice of most interventional radiologists, where the ideal liquid embolic agent is far from being identified. Non-adhesive liquid embolic agents (NALEA) harden from the outside in, resulting in deep penetration, known as a “magma-like” progression, which allows for more distal embolization with good control of the embolic material. The present multicenter retrospective cohort study evaluated the efficacy, feasibility, and safety of transcatheter arterial embolization (TAE) with ethylene-vinyl alcohol (EVOH)-based NALEAs (Onyx and Squid) in acute bleeding outside the neurovascular area.

I consider the tópic relevant and original in the field. The subject still lacks scientific literature that proves the methods for this purpose and therefore, the present article sheds light on a serious medical subject and is still without fully defined solutions.

The article presents the methodology for treating non-neurovascular vascular bleeding with non-adherent embolic fluids and a report on a large number of patients treated. Thus, I consider that the data presented complement the available literature.

There are no improvements to be added to the methodology. Unfortunately, due to the seriousness of the cases presented, it is not possible to carry out a controlled study.

The conclusions are consistent with the main objectives, evidence, and arguments presented.

The authors carried out an extensive literature review, including very up-to-date articles in good numbers. I consider it appropriate in this regard.

I found the figures to be very representative, very beautiful and appropriate. The tables are correctly presented and with well-constructed content.

As a suggestion, in the abstract, on page 1, lines 33 and 34, please remove the study inclusion period already described above.

Author Response

Dear Reviewer,

Thanks for your comments.

The abstract has been updated as you suggested.

Best regards,

Roberto Minici